# Crystalline Orientation-Dependent Ferromagnetism in N^+^-Implanted MgO Single Crystal

**DOI:** 10.3390/ma15207274

**Published:** 2022-10-18

**Authors:** Xingyu Wang, Chunlin Ma, Weiping Zhou, Weishi Tan

**Affiliations:** 1School of Science, Nanjing University of Science and Technology, Nanjing 210094, China; 2School of Physics and Electronic Electrical Engineering, Huaiyin Normal University, Huaian 223001, China; 3School of Materials Science and Engineering, Nanchang University, Nanchang 330031, China; 4All-Solid-State Energy Storage Materials and Devices Key Laboratory of Hunan Province, College of Information and Electronic Engineering, Hunan City University, Yiyang 413002, China

**Keywords:** N-implanted MgO, orientation-dependent, defect-induced ferromagnetism, microstructure, spintronic

## Abstract

Samples of (110), (100), and (111) MgO single crystals were implanted with 70 keV N ions at room temperature. All as-implanted samples showed room temperature hysteresis in magnetization loops. The observed saturation magnetization (*Ms*) was 0.79 × 10^−4^ emu/g, 1.28 × 10^−4^ emu/g, and 1.5 × 10^−4^ emu/g for (110), (100) and (111) orientation implanted-MgO and follows the relation *Ms*(111) > *Ms*(100) > *Ms*(110), indicative of crystalline orientation-dependent ferromagnetism in N-implanted MgO. The samples were characterized by X-ray photoelectron spectroscopy (XPS), high resolution X-ray diffraction (HRXRD), reciprocal space mapping (RSM), and photoluminescence (PL). The results indicated that the amount of N-substitute-O and N-interstitial defects in these three N-implanted MgO samples showed the same changing tendency as compared with *Ms* data. Thus, we conclude that the N-substitute-O and N-interstitial defects may play a crucial role in controlling the N^+^-implanted-induced ferromagnetism.

## 1. Introduction

Due to the prospect of fabricating faster and more energy-efficient circuits than standard semiconductor devices, spintronics have attracted continuous attention in the past several decades. Some research studies have reported novel achievements that can provide new forms of functionality, such as spin field-effect transistors, giant magnetoresistive sensors, storage media, etc. [1,2,3]. The critical factor for realizing spintronic applications is to produce ferromagnetism at room temperature (RTFM) in semiconductors. Initially, researchers focused on the diluted magnetic semiconductor (DMS), which can obtain magnetic moments by doping transition-mental elements [4]. In fact, RTFM has been observed in a significant amount of DMS materials [5]. However, some unsolved difficulties for DMSs, such as non-uniform spin density and magnetic impurity clusters, still present obstacles to realizing the spintronic devices [6].

To overcome these limitations, the RTFM was improved by doping nonmagnetic elements in traditional semiconductors [7]. Along this way, a series of theoretical and experimental investigations have been carried out to exploit suitable spintronic devices. Researchers have recently focused on the RTFM in metal oxide semiconductors doped with nonmagnetic components, such as aluminum-doped TiO_2_ [8], Ag-doped ZnO [9], and Li-doped SnO_2_ nanoparticles [10], etc. All of the studies concluded that the induced magnetic moment in these materials is attributed to the diverse defects. However, the relationship between the induced ferromagnetism and the microstructure of the materials has not been well established, and specific details of the defect-mediated mechanism(s) are still obscure. This hinders the further development of spintronic devices. Usually, several methods, including nano-scale [11], doping concentration [12], and annealing effect [13], are used to investigate the correlation between the produced defects and the induced ferromagnetism. However, the crystallographic orientation’s influence on defect-induced ferromagnetism has been neglected, although it is a more intuitive way to investigate the effect of crystal defects on magnetic properties. 

Magnesium oxide (MgO) is considered to be a promising material in the field of spintronic devices due to its wide band gap, optical behavior, and multilevel switching characteristics [14]. Therefore, MgO single crystals have been intensively studied in the past decade in terms of spintronic devices. Some previous papers reported the defect-induced RTFM in MgO [15,16]. Noticeably, MgO single crystal exhibits a rock-salt structure. The charge and atomic density distribution differs on different crystallographic planes, which means that the implantation process will present an orientation-dependent characteristic in this material. Several previous studies in the literature have reported materials’ orientation-dependent physical and chemical properties [17,18,19].

Ion implantation is a promising tool in the field of defect engineering due to its controllable, reproducible, efficient, and non-equilibrium characteristics. The implanted ions may change the crystal structural properties and electrical properties due to the energetic collision cascades, and endow unexpected properties to the materials. In the case of MgO, numerous experiments on ion implantation have been done. G. Baubekov et al. [20] deeply investigated the fluence-dependence concentration of F-type defects and the micro-mechanical radiation effect in Xe-implanted MgO crystal. Eugene Kotomin et al. [21] revealed that the defect migration parameters have fluence-dependent characteristics through a systematic literature analysis. Additionally, research studies have demonstrated that ion implantation is a feasible approach to obtain ferromagnetism in MgO [14].

In the current paper, we implanted N ions into MgO single crystals with (100), (110), and (111) orientations and successfully observed orientation-dependent ferromagnetism. To clarify the orientation-dependent ferromagnetism, the microstructures of the pristine and as-implanted samples were characterized in detail. Our results indicate that defect type was crucial in introducing ferromagnetism in MgO. The discovery may provide a valuable reference for controlling ferromagnetism in oxide semiconductors.

## 2. Experimental Details

MgO single crystals with (100), (110), and (111) orientations were purchased from Hefei Kejing Materials Technology Co., Ltd. (Hefei, China). The samples were prepared using the arc melting method. The samples were cut into the size of 10 mm × 5 mm × 0.5 mm. An ion implantation experiment was conducted at the Institute of Semiconductors, CAS, Beijing. N ions with the energy of 70 keV were implanted into MgO samples with the influent of 2 × 10^17^ cm^−2^ at room temperature. Samples were placed on a wafer tray during the process of ion implantation to avoid contact with metal support. The implantater was performed at an angle of 7° off normal direction to decrease the channeling impact. The base pressure of the chamber was maintained at 2 × 10^−4^ Pa. Qualitative analysis of the elements experiment was carried out by X-ray photoelectron spectroscopy (XPS, PHI Quantera) with monochromatic Al Kα X-ray radiation. The magnetic measurements data were obtained through a superconducting quantum interference device (SQUID, Quantum Design) at room temperature. The applied magnetic field was parallel to the sample surface. Both before and during the process of magnetic measurement, the operation was conducted carefully to avoid magnetic impurities contamination. The microstructures of pristine and as-implanted samples were characterized in Beijing Synchrotron Radiation Facility (BSRF) with the X-ray wavelength of 0.15493 nm. The high-resolution X-ray diffraction (HRXRD) and reciprocal space mapping (RSM) were measured with a scanning step width of 0.02°. The photoluminescence (PL) emission spectra were measured by spectrofluorometer (FS5, Edinburgh Instruments, Edinburgh, UK) using the 370 nm laser diode (LD) as an excitation source.

## 3. Result and Discussion

The magnetic field-dependent magnetization (M-H) data were measured at room temperature to investigate these samples’ magnetic properties. Figure 1 shows the M-H curves of the pristine and as-implanted MgO samples with different crystalline orientations. The diamagnetic background signals have been subtracted. The pristine MgO displays non-ferromagnetism properties. However, all the implanted samples present well-defined hysteresis behaviors. The coercive field of the as-implanted samples with (110), (100), and (111) orientations are approximately 78 Oe, 83 Oe, and 95 Oe, respectively. The result indicates that the magnetic behavior of MgO is changed by N ions implantation, which is consistent with a previous study [22]. Moreover, an interesting phenomenon is that saturation magnetization (*Ms*) shows apparent crystallographic orientation dependence. The magnitude of saturation magnetization of as-implanted samples with different orientations is clearly presented in Figure 1d. The saturation magnetization (*Ms*) in as-implanted samples with (111) orientation is the largest, at about 1.5 × 10^−4^ emu/g, and the smallest saturation magnetization emerged in the sample with (110) orientation, at about 0.79 × 10^−4^ emu/g, which is twice as small as the largest one. Obviously, such a large difference cannot be a measurement error. Normally, several factors could influence ferromagnetism in doped systems, including crystallization, doped environment, and implantation dose, etc. In our experiment, the above variables were unified. Theoretical and experimental research studies propose that the magnitude of saturation magnetization is determined by the defect concentration [23], but there is some disagreement regarding which defect types can truly induce ferromagnetism. Generally, there are three charge state for Mg vacancy (V_Mg_^0^, V_Mg_^−^, V_Mg_^2−^) and O vacancy (V_O_^0^, V_O_^+^, V_O_^2+^). A review article [24] summarized the magnetic properties induced by Mg and O vacancy in these three charge states: V_Mg_^0^ and V_Mg_^−^ are responsible for the magnetic moment, while the V_Mg_^2−^, V_O_^0^, V_O_^+^ and V_O_^2+^ have no contribution to the magnetic moment. Furthermore, a previous study demonstrated that the amount of displaced atoms and damage in Ar^+^-implanted MgO depends on the crystallographic orientation [25]. The above case also occurred in N^+^-implanted MgO. The amounts of displaced atoms increased from (111) through (100) to the (110) orientation, which is consistent with the rising tendency of *Ms* in different orientations. Such a difference in the number of displaced atoms should result in the concentration difference of specific defect types. Based on the above results, we speculate that the number of certain defect types may be a factor in explaining the observed orientation-dependent ferromagnetism.

The XPS was carried out to clarify the difference in defect composition in these pristine and implanted samples. In spite of the magnetic measurement process being conducted carefully, the XPS was still used to exclude possible extraneous magnetic impurities, such as Fe, Co, Ni, Mn, and Cr. The XPS full spectrum is shown in Figure 2a–c. No ferromagnetic impurities peaks are detected, which is indicative of the absence of any magnetic contamination. Previous calculations suggest that the intrinsic defects can barely induce ferromagnetism in bulk MgO [26]. Thus, we focus on the extrinsic N-related defects. As displayed in Figure 2d–f, there emerges an asymmetric N 1s peak between 395 eV and 401 eV in all of the N^+^-implanted samples. All the N 1s peaks of the N^+^-implanted samples can be fitted into two symmetric peaks, with N_a_ located at 398.9 eV~399 eV and N_b_ located at about 398.8~399.1 eV. The N_a_ can be obviously assigned to α-N_2_ [27]. Referring to the literature [14], the N_b_ peak can be ascribed to N-related defects. Noticeably, the relative peak area of N_b_ increases from (110) through (100) to the (111) orientation, and the value is 41%, 56.8%, and 62.1%, respectively. The relative peak area of N_b_ is positively associated with the magnitude of the magnetic moment. Hence, we conclude that the N-related defects play a key role in inducing the ferromagnetism in N^+^-implanted MgO. The most possible situation is that N ions substitute O ions (N_O_) or N ions locate at the interstitial site (N_int_) even though the existence of vacancy-Mg and vacancy-O and Interstitial-O are prior to N_O_ and N_int_ in terms of formation energy [26]. The unpaired electronic coupling between these defects may change the magnetic properties.

To further investigate the defect types in as-implanted MgO, we measured RSMs of 200, 220, and 111 reciprocal lattice points of the pristine and as-implanted MgO samples with (100), (110), and (111) orientation, respectively. The results are shown in Figure 3, in which q_y_ and q_z_ are the horizontal and vertical component of the scattering vector, respectively. According to diffuse scattering theory, defects can give rise to lattice distortions, and the long-range part of this distortion leads to the scattering vector deviating from the reciprocal lattice points [28]. Simultaneously, this lattice distortion generates diffuse scattering centered on the reciprocal lattice point, which can be visually displayed in RSM. The previous simulation exhibited that different defect configurations can produce different diffuse-scattering distribution shapes [29]. Figure 3 shows the RSMs of pristine and implanted MgO samples. All the implanted samples demonstrate obvious diffuse scattering in comparison with pristine samples, indicating the introduction of various defects. However, as is shown in Figure 3a,d, before and after N^+^-implantation, the shape of scattering diffuse distribution has no significant change, which suggests that the intrinsic defects are still dominate. On the other hand, for Figure 3b,c,e,f, the distribution shape changes a lot. These results indicate that the dominant defect types are dissimilar among the (110), (100), and (111) orientation implanted samples. This means that higher concentrations of some kind of defect types have been introduced. Some researchers propose that the diffuse scattering along radial directions of the reciprocal space is sensitive to defect types, and the defect types that can cause lattice expansion may lead to higher diffuse scattering intensity [17,30]. The scattering intensity of N^+^-implanted (110), (100), and (111) orientation samples along the q_y_-axis are displayed in Figure 3g–i, respectively. We can observe that the scattering intensity and distribution area shows an increasing tendency from (110) to (100) to (111) as-implanted samples, which means that the degree of lattice distortion caused by defects is strongest in the (111) orientation as-implanted sample, while weakest in the (110) orientation as-implanted sample. Therefore, we infer that the concentration of defect types which can cause lattice expansion should adhere to the following changing tendency: (111) > (100) > (110).

In order to get more defect-related information, high-resolution X-ray diffraction was used to characterize the microstructure. Figure 4a–c shows the HRXRD pattern of pristine and N^+^-implanted samples. As shown in the inset, all samples present a single crystal structure, since no additional second-phase diffraction peaks were detected. Additionally, the diffraction intensity of all N^+^-implanted samples decreases compared to the pristine samples, which is indicative of the lattice disorder resulting from the implantation-induced defects. Remarkably, all the diffraction peaks move to a lower angle position in as-implanted samples compared with the pristine MgO. The value of peak shift increases from (110) to (100) to the (111) orientation by 0.078°, 0.237°, and 0.278°, respectively. The peak left-shift is related to the lattice expansion. In MgO, the lattice expansion likely originates from N_int_ or N_O_ [31]. As is known, the XRD result is an average effect of the crystalline field. Thus, the value of peak shift is proportional to the amount of N_int_ and N_O_ defects. This result is consistent with the RSM results.

The PL spectra were measured at room temperature. As shown in Figure 5a, there is an apparent emission band located at approximately 2.48~3.1 eV for all the as-implanted samples. Uchino and Okutsu [32] reported that two broad PL bands at 2.48~3.1 eV were excited in a MgO microcrystal. By means of analyzing the variation of emission peak before and after annealing, they concluded that the PL peaks were related to oxygen vacancies. We used their approach for reference. The PL spectra of the as-implanted sample before and after annealing in air at 400 ℃ for 10 min were measured. As displayed in Figure 5b, the PL emission intensity obviously decreases. Meanwhile, the shape of the excited peak at 2.8 eV (440 nm) changes. Thus, we confirm that the broad peak located at 2.48~3.1 eV should be related to oxygen vacancies. For the purpose of further exploring the defect types, we decomposed the emission spectrum through Gaussian Fit. Since the emission peaks of the (110), (100), and (111) orientations have a similar shape, we just selected the peak of the (111) implanted sample for decomposition. As is shown in Figure 5b, the emission spectrum can be decomposed into three Gaussian peaks located at 3 eV (410 nm), 2.83 eV (438 nm), and 2.64 eV (469 nm). We can assign the emission band observed at 3 eV to F^+^ (oxygen vacancy with one electron) center, which is originated from the ^2^T_1u_ → ^2^A_1g_ transition [33]. The emission peak located at 2.83 eV is owing to ^3^B_1u_ → ^1^Ag transitions of the F_2_^2+^ (combinations of neighboring F centers with two electrons) center [34]. The emission band appearing at 2.64 eV can be attributed to F type anion vacancies [35]. However, results shown in Figure 5a indicate that from the as-implanted samples with the orientation of (110) to (100) to (111), the luminescent intensity decreases progressively. The reason for this phenomenon is that, in the process of implantation, a large number of N-interstitial and O-vacancy defects are introduced, and in subsequent thermal equilibrium, the N-interstitial occupies the site of O-vacancies. The PL emission intensity suggests that the as-implanted sample with (110) orientation has the largest number of O-vacancies, while the sample with (111) orientation holds the largest amount of N-interstitial and N-substitute-O defects. This result is in good agreement with the XPS results. Furthermore, when considering the magnetic measurement results, we can conclude that the magnitude of the magnetic moment is independent of the O-vacancies defect concentration, while the N-related defects have a positive effect.

Based on the above experimental results and literature description, we can draw the conclusion that the concentration of N_int_ and N_O_ defects are orientation-dependent in N^+^-implanted MgO single crystals. The concentration of N-related defects decreases from (111) through (100) to (110) N^+^-implanted MgO. This is probably due to the distinctive properties in the different crystallographic planes. The charge and atomic density distributions existing on the (110), (110), and (111) planes are dissimilar. Usov et al. [36] reported that the (110) orientation is the most damage-resistant orientation compared to (100) and (111) in MgO single crystal, and following the rule is (111) > (100) > (110). Therefore, the cascade collision and secondary collision occurring on different crystallographic planes should cause different degrees of damage for the crystal structure, thereby affecting the concentration and species of defects. Furthermore, the efficiency of the mobility of point defects also depends on the sample crystallographic orientation during irradiation [37]. Mao et al. [38] investigated Fe-implanted MgO. They found that the Fe ions were easiest to migrate inside the crystal on the (111) plane, while it was hardest on the (110) plane. Accordingly, for the case of N^+^-implanted MgO, the N-related defects are more likely generated in (111) and (100) oriented samples than in the (110) oriented sample due to the weaker damage resistance and higher defect migration efficiency.

The observed value of saturation magnetization is subject to the same changing tendency of N-related concentration. This result demonstrates that the N-interstitial and N-substitute-O defects introduce ferromagnetism and enhance the magnetic moment. Both of these two defect types can contribute unpaired electrons from N 2p orbitals, and produce the localized spin magnetic moment. Moreover, according to the Heisenberg exchange model, the coupling between the unpaired electrons from the N-related defects and intrinsic vacancies can result in local magnetic ordering through exchange correlation. This should be the source of ferromagnetism.

## 4. Conclusions

In summary, apparent RTFM was observed in N^+^-implanted MgO single crystals with (100), (110), and (111) orientation. The saturation magnetization presents an orientation-dependent characteristic. The RSM, XPS, HRHRD, and PL results demonstrated that the concentration of N-substitute-O and N-interstitial defects has a similar orientation-dependent characteristic with the magnetic properties. The orientation-dependent concentration of N-related defects may be due to the distinctive properties in the different crystallographic planes. Consequently, the localized magnetic moments induced by N-related defects were also discussed.

## Figures and Tables

**Figure 1 materials-15-07274-f001:**
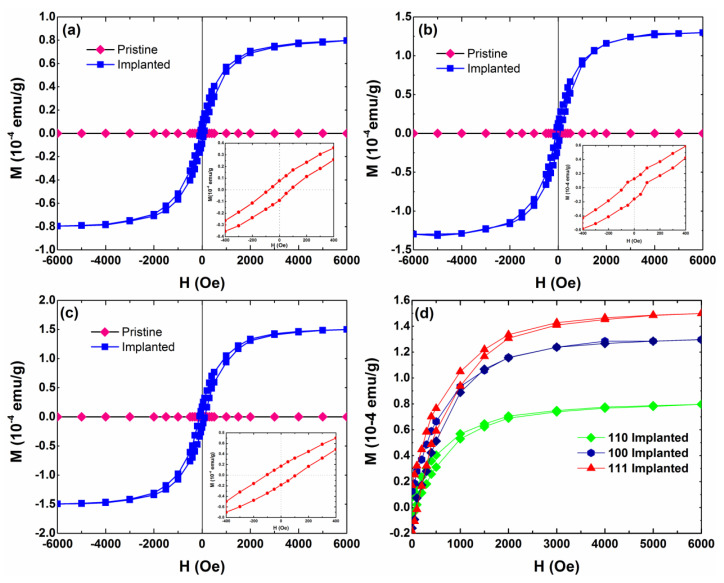
Ferromagnetic hysteresis loops at room temperature for pristine and N^+^-implanted MgO samples with (**a**) (110) orientation, (**b**) (100) orientation, and (**c**) (111) orientation. The implantation dose of all samples is 2 × 10^17^ ions cm^−2^; (**d**) The magnitude of saturation magnetization of MgO samples with the three orientations. The insets in (**a**–**c**) display a magnified view of the low field data.

**Figure 2 materials-15-07274-f002:**
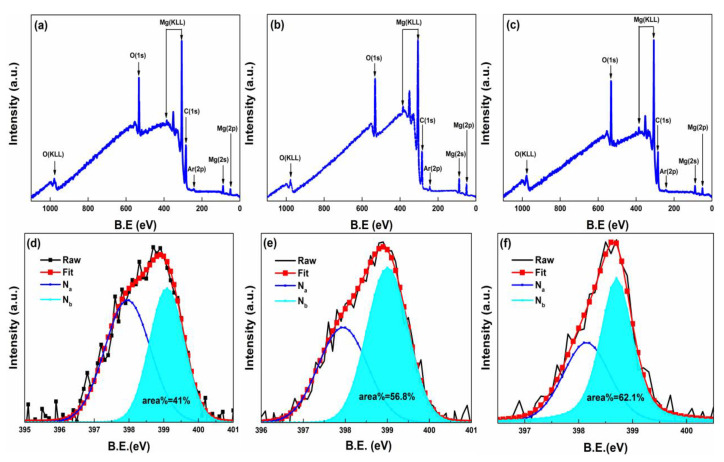
Total XPS spectrum of the pristine samples with the (**a**) (110) orientation, (**b**) (100) orientation, and (**c**) (111) orientation. N 1s XPS signals of as-implanted samples with (**d**) (110), (**e**) (100), and (**f**) (111) orientation.

**Figure 3 materials-15-07274-f003:**
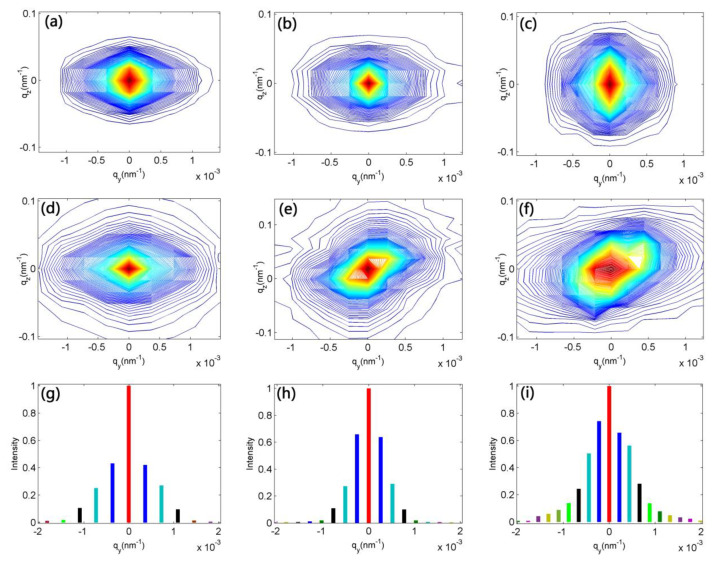
RSM of pristine and as-implanted MgO samples: (**a**,**d**) (110) orientation; (**b**,**e**) (100) orientation; (**c**,**f**): (111) orientation; (**g**–**i**): diffuse scattering distribution of implanted (110), (100), and (111) samples, respectively.

**Figure 4 materials-15-07274-f004:**
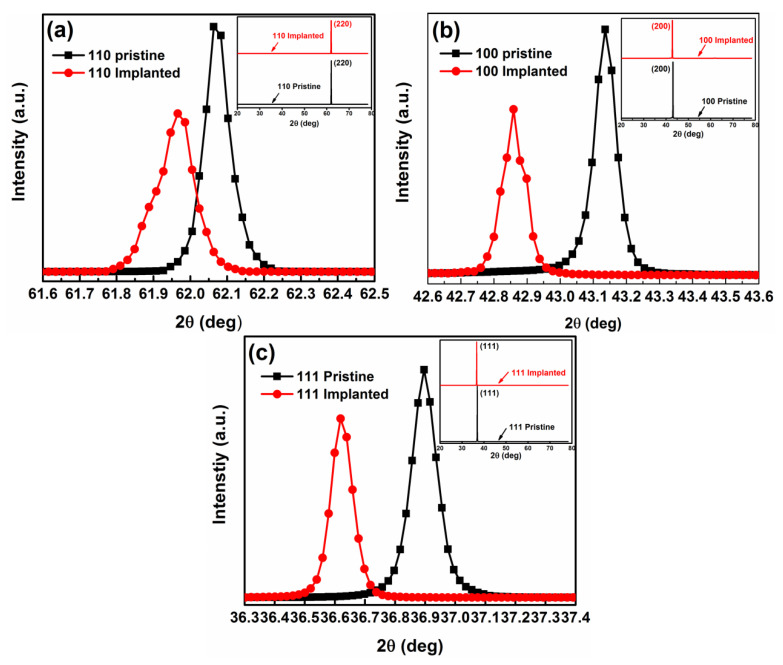
HRXRD patterns of for pristine and implanted MgO samples with (**a**) (110) orientation, (**b**) (100) orientation, and (**c**) (111) orientation.

**Figure 5 materials-15-07274-f005:**
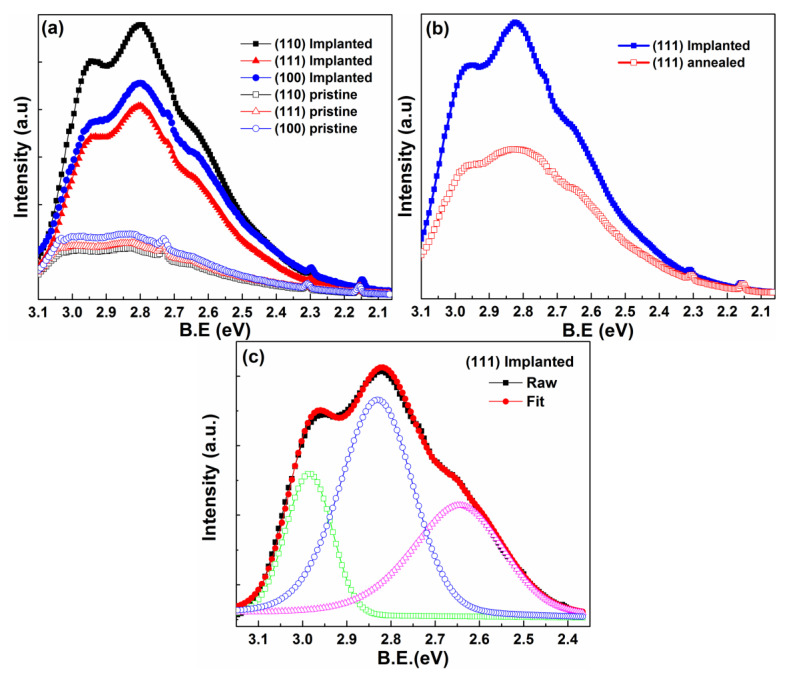
(**a**) PL of as-implanted samples with different orientation. (**b**) PL of as-implanted sample before and after annealing in air. (**c**) Gaussian fit of PL spectra for (111) implanted MgO sample.

## Data Availability

Not applicable.

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
