# Peer review of "Crystalline Orientation-Dependent Ferromagnetism in N+-Implanted MgO Single Crystal"

_materials, 2022, doi:10.3390/ma15207274_

Round 1
Reviewer 1 Report
Refereree report on manuscript “Crystalline orientation dependent ferromagnetism in N-implanted MgO single crystal”
This version does not look worthy and cannot be recommended for publication in this form and at least needs major revision.
1. Since the effects associated with irradiation of MgO are included in this work, it is absolutely necessary to add an additional paragraph on radiation phenomena and radiation defects in MgO to the Introduction. The very recent works related to MgO include:
Baubekova, G., et al (2020). Accumulation of radiation defects and modification of micromechanical properties under MgO crystal irradiation with swift 132Xe ions. Nuclear Instruments and Methods in Physics Research Section B: Beam Interactions with Materials and Atoms, 463, 50-54.
See also, a detailed analysis of many different irradiation experiments in:
Kotomin, E., et al (2018). Anomalous kinetics of diffusion-controlled defect annealing in irradiated ionic solids. The Journal of Physical Chemistry A, 122(1), 28-32.
2. In experimental details, please justify the choice of using a 370 nm laser diode (LD) as the excitation source for photoluminescence.
3. Authors should remember that both cationic and anionic vacancies in MgO are in three charge states. This needs to be emphasized more clearly.
4. PL part of the paper. It is important to note here that a 370 nm excitation source is not sufficient to excite oxygen vacancies (F and F+ centers - oxygen vacancy with two or one electrons), since they have extinction bands at 250 nm. See detailed, synchrotron luminescence spectroscopy study: Popov, A. I., et al. "Comparative study of the luminescence properties of macro-and nanocrystalline MgO using synchrotron radiation." Nuclear Instruments and Methods in Physics Research Section B: Beam Interactions with Materials and Atoms 310 (2013): 23-26.
5. The data in Figure 5 is not complete. The spectra must be decomposed into individual Gaussians when the horizontal axis is in the energy scale (eV or cm-1). After which the individual bands must be identified and the questions why they are excited at 370 nm must be answered.
In general, the manuscript is interesting and can be recommended for publication after constructive reflection on the above comments.
Author Response
Dear Reviewer,
Thank you very much for your recommendation.
We have tried our best to revise the manuscript according to your kind and construction comments and suggestions. We sincerely hope that this revised manuscript has addressed all your comments and suggestions.
Please find my itemized responses and revisions in the re-submitted files
Yours Sincerely,
Xingyu

Reviewer 2 Report
Wang and coauthors report magnetic properties of N-implanted oriented crystals of MgO. They show that, depending on the orientation of MgO crystals, the concentration of N-induced defects is different, which results in different values of saturation magnetization. Also, the authors discuss the nature of defects on the basis of various results including luminescent spectroscopy, high-resolution X-ray diffraction, and X-ray photoelectron spectroscopy. The paper is quite important as it presents a new step towards materials for spintronics; in general, it is well-written, and the discussion is concise. In particular, I appreciate that the authors present some of their results as hypotheses, if the absolute prove is still impossible. I recommend publishing this paper after minor revision.
1. Figure 4 shows that after implantation the width of the (200) peak has increased only in the case of the [111] orientation. Why? Does it mean the distortion of the crystal lattice along with its expansion?
2. It is necessary to check the grammar. For instance, (page 4, line 4) “The XPS full spectrum are shown”, (page 5, line 2 from bottom) “As shown inset, all samples present a single…”, or (page 6, line 7 from bottom) “the PL peaks was related…”. Please double-check!
Author Response
Dear Reviewer,
Thank you very much for your recommendation.
We have tried our best to revise the manuscript according to your kind and construction comments and suggestions. We sincerely hope that this revised manuscript has addressed all your comments and suggestions.
Please find my itemized responses and revisions in the re-submitted files.
Yours Sincerely,
Xingyu

Reviewer 3 Report
This is an interesting topic and deserves careful investigation. However, the only direct measurement of the magnetism is via SQUID magnetometry and there is a long, troubled history in DMS field of reporting extrinsic impurities/clusters (as authors point out in intro) and claiming it is intrinsic.
1) in Fig 1 they do NOT show the pristine sample, this must be shown after slope subtraction to prove that there is no FM impurity present.
2) the handling of samples -- mounting, unmounting, steel tweezers, etc. -- can all introduce magnetic impurities. We should take great care to exclude trivial sources of magnetism by singling out the effect of N implantation. One way is to make a dose test (1x, 10x) and even have a sample go through the whole process without getting any dose and showing that the magnetism correlates with N dose.
3) XPS was stated to not find any FM impurities, however this technique is only sensitive to 0.1 at% and the level of impurities that would explain your 10^-5 emu signal (10^-4 emu/g * 0.1 g sample) is in the ppm or below.
4) in abstract, on line 3 the value of Ms for (100) crystal was dropped.
Author Response

(The authors gave the same response as above.)

Round 2
Reviewer 1 Report
The authors have strongly improved their original manuscript, which now can be accepted for publication
Reviewer 3 Report
thank you for addressing these points about magnetic signals, i feel that you have done what you can to correlate magnetism with N+ implantation and i feel this is ready for publication.